# Differentiable Segmentation of Sequences

**Erik Scharwächter**[*]
TU Dortmund University, Germany

**Jonathan Lennartz**
University of Bonn, Germany

**Emmanuel Müller**
TU Dortmund University, Germany

## Abstract

Segmented models are widely used to describe non-stationary sequential data with discrete change points. Their estimation usually requires solving a mixed discrete-continuous optimization problem, where the segmentation is the discrete part and all other model parameters are continuous. A number of estimation algorithms have been developed that are highly specialized for their specific model assumptions. The dependence on non-standard algorithms makes it hard to integrate segmented models in state-of-the-art deep learning architectures that critically depend on gradient-based optimization techniques. In this work, we formulate a relaxed variant of segmented models that enables joint estimation of all model parameters, including the segmentation, with gradient descent. We build on recent advances in learning continuous warping functions and propose a novel family of warping functions based on the two-sided power (TSP) distribution. TSP-based warping functions are differentiable, have simple closed-form expressions, and can represent segmentation functions exactly. Our formulation includes the important class of segmented generalized linear models as a special case, which makes it highly versatile. We use our approach to model the spread of COVID-19 with Poisson regression, apply it on a change point detection task, and learn classification models with concept drift. The experiments show that our approach effectively solves all these tasks with a standard algorithm for gradient descent.

## 1 Introduction

Non-stationarity is a classical challenge in the analysis of sequential data. A common source of non-stationarity is the presence of change points, where the data-generating process switches its dynamics from one regime to another regime. In some applications, the *detection* of change points is of primary interest, since they may indicate important events in the data (Page, 1954; Box & Tiao, 1965; Basseville & Nikiforov, 1986; Matteson & James, 2014; Li et al., 2015; Arlot et al., 2019; Scharwächter & Müller, 2020). Other applications require *models* for the dynamics within each segment, which may yield more insights into the phenomenon under study and enable predictions. A plethora of segmented models for regression analysis (McGee & Carleton, 1970; Hawkins, 1976; Lerman, 1980; Bai & Perron, 2003; Muggeo, 2003; Acharya et al., 2016) and time series analysis (Hamilton, 1990; Davis et al., 2006; Aue & Horváth, 2013; Ding et al., 2016) have been proposed in the literature, where the segmentation materializes either in the data dimensions or the index set.

We adhere to the latter approach and consider models of the following form. Let $x = (x_1, ..., x_T)$ be a sequence of $T$ observations, and let $z = (z_1, ..., z_T)$ be an additional sequence of covariates used to predict these observations. Observations and covariates may be scalars or vector-valued. We refer to the index $t = 1, ..., T$ as the *time of observation*. The data-generating process (DGP) of $x$ given $z$ is *time-varying* and follows a segmented model with $K \ll T$ segments on the time axis. Let $\tau_k$ denote the beginning of segment $k$. We assume that

$$x_t \mid z_t \overset{\text{iid}}{\sim} f_{\text{DGP}}(z_t, \theta_k), \text{ if } \tau_k \leq t < \tau_{k+1}, \tag{1}$$

---

[*]corresponding author, e-mail: `erik.scharwaechter@cs.tu-dortmund.de`

where the DGP in segment $k$ is parametrized by $\theta_k$. This scenario is typically studied for change point detection (Truong et al., 2020; van den Burg & Williams, 2020) and modeling of non-stationary time series (Guralnik & Srivastava, 1999; Cai, 1994; Kohlmorgen & Lemm, 2001; Davis et al., 2006; Robinson & Hartemink, 2008; Saeedi et al., 2016), but also captures classification models with concept drift (Gama et al., 2013). and segmented generalized linear models (Muggeo, 2003)

We express the segmentation of the time axis by a segmentation function $\zeta : \{1, ..., T\} \longrightarrow \{1, ..., K\}$ that maps each time point $t$ to a segment identifier $k$. The segmentation function is monotonically increasing with boundary constraints $\zeta(1) = 1$ and $\zeta(T) = K$. We denote all segment-wise parameters by $\theta = (\theta_1, ..., \theta_K)$. The ultimate goal is to find a segmentation $\zeta$ as well as segment-wise parameters $\theta$ that minimize a loss function $\mathcal{L}(\zeta, \theta)$, for example, the negative log-likelihood of the observations $x$. Existing approaches exploit the fact that model estimation within a segment is often straightforward when the segmentation is known. These approaches decouple the search for an optimal segmentation $\zeta$ algorithmically from the estimation of the segment-wise parameters $\theta$:

$$\min_{\zeta,\theta} \mathcal{L}(\zeta, \theta) = \min_{\zeta} \min_{\theta} \mathcal{L}(\zeta, \theta). \tag{2}$$

Various algorithmic search strategies have been explored for the outer minimization of $\zeta$, including grid search (Lerman, 1980), dynamic programming (Hawkins, 1976; Bai & Perron, 2003), hierarchical clustering (McGee & Carleton, 1970) and other greedy algorithms (Acharya et al., 2016), some of which come with provable optimality guarantees. These algorithms are often tailored to a specific class of models like piecewise linear regression, and do not generalize beyond. Moreover, the use of non-standard optimization techniques in the outer minimization hinders the integration of such models with deep learning architectures, which usually perform joint optimization of all model parameters with gradient descent.

In this work, we provide a continuous and differentiable relaxation of the segmented model from Equation 1 that allows joint optimization of all model parameters, including the segmentation function, using state-of-the-art gradient descent algorithms. Our formulation is inspired by the learnable warping functions proposed recently for sequence alignment (Lohit et al., 2019; Weber et al., 2019). In a nutshell, we replace the hard segmentation function $\zeta$ with a soft warping function $\gamma$. An optimal segmentation can be found by optimizing the parameters of the warping function. We propose a novel class of piecewise-constant warping functions based on the two-sided power (TSP) distribution (Van Dorp & Kotz, 2002; Kotz & van Dorp, 2004) that can represent segmentation functions exactly. TSP-based warping functions are differentiable, have simple closed-form expressions that allow fast evaluation, and their parameters have a one-to-one correspondence with segment boundaries. Source codes for the model and all experiments can be found in the online supplementary material at `https://github.com/diozaka/diffseg`.

## 2 RELAXED SEGMENTED MODELS

We relax the model definition from Equation 1 in the following way. We assume that

$$x_t \mid z_t \overset{\text{iid}}{\sim} f_{\text{DGP}}\left(z_t, \hat{\theta}_t\right), \tag{3}$$

where we substitute the actual parameter $\theta_k$ of the DGP at time step $t$ in segment $k$ by the predictor $\hat{\theta}_t$. The predictor $\hat{\theta}_t$ is a weighted sum over the individual segment parameters

$$\hat{\theta}_t := \sum_k \hat{w}_{kt} \theta_k \tag{4}$$

with weights $\hat{w}_{kt} \in [0, 1]$ such that $\sum_k \hat{w}_{kt} = 1$ for all $t$. The weights can be viewed as the entries of a (soft) alignment matrix that aligns every time step $t$ to a segment identifier $k$. Given a (hard) segmentation function $\zeta$, we can construct a (hard) alignment matrix by setting $\hat{w}_{kt} = 1$ if and only if $\zeta(t) = k$. In our relaxed model, we employ a continuous predictor $\hat{\zeta}_t \in [1, K]$ for the value of the segmentation function $\zeta(t)$ and let the weight $\hat{w}_{kt}$ depend on the difference between $\hat{\zeta}_t$ and $k$:

$$\hat{w}_{kt} := \max\left(0, 1 - \left|\hat{\zeta}_t - k\right|\right) \tag{5}$$

The smaller the difference, the closer the alignment weight $\hat{w}_{kt}$ will be to 1. If $k \le \hat{\zeta}_t \le k + 1$, this choice of weights leads to a linear interpolation of the parameters $\theta_k$ and $\theta_{k+1}$ in Equation 4. Higher-order interpolations can be achieved by adapting the weight function accordingly.

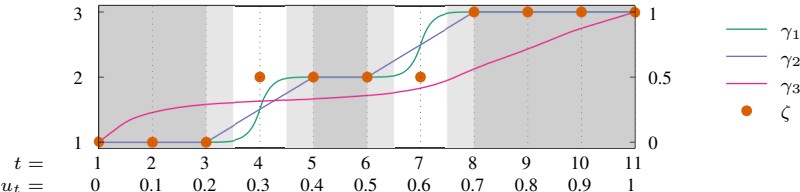

Figure 1: Example segmentation function $\zeta(t)$ and warping functions $\gamma_i(u)$. The shaded regions are piecewise constant in $\gamma_1$ and $\gamma_2$, respectively; $\gamma_3$ is strictly increasing.

The key question is how to effectively parametrize the continuous predictors $\hat{\zeta}_t$ for the segmentation function. For this purpose, we observe that the continuous analogue of a monotonically increasing segmentation function is a warping function (Ramsay & Li, 1998). Warping functions describe monotonic alignments between closed continuous intervals. Formally, the function $\gamma : [0,1] \longrightarrow [0,1]$ is a warping function if it is monotonically increasing and satisfies the boundary constraints $\gamma(0) = 0$ and $\gamma(1) = 1$. Any warping function $\gamma$ can be transformed into a continuous predictor $\hat{\zeta}_t$ by evaluating $\gamma$ at $T$ evenly-spaced grid points on the unit interval $[0,1]$ and rescaling the result to the domain $[1,K]$. Let $u_t = (t-1)/(T-1)$ for $t = 1,...,T$ be a unit grid. We define

$$\hat{\zeta}_t := 1 + \gamma(u_t) \cdot (K-1). \tag{6}$$

Similar transformations of warping functions have recently been applied for time series alignment with neural networks (Weber et al., 2019). An example segmentation function and predictors based on several warping functions are visualized in Figure 1. With the transformation from Equation 6, our relaxed segmented model is parametrized by the segment parameters $\theta$ and the parameters of the warping function $\gamma$ that approximates the segmentation. The discrete-continuous optimization problem from Equation 2 has changed to a continuous optimization problem that is amenable to gradient-based learning:

$$\min_{\gamma,\theta} \mathcal{L}(\gamma,\theta). \tag{7}$$

Figure 1 illustrates that the ideal warping function for segmentation is piecewise constant—a formal argument is provided in Appendix A. Several families of warping functions have been proposed in the literature and can be employed within our model (Aikawa, 1991; Ramsay & Li, 1998; Gervini & Gasser, 2004; Gaffney & Smyth, 2004; Claeskens et al., 2010; Freifeld et al., 2015; Detlefsen et al., 2018; Weber et al., 2019; Kurtek et al., 2011; Lohit et al., 2019). However, none of them contains piecewise-constant functions, and many of them are more expressive than necessary for the segmentation task. Below, we define a novel family of piecewise-constant warping functions that is tailored specifically for the segmentation task, with only one parameter per change point.

## 3 TSP-BASED WARPING FUNCTIONS

Warping functions are similar to cumulative distribution functions (cdfs) for random variables (Lohit et al., 2019). Cdfs are monotonically increasing, right-continuous, and normalized over their domain (Wasserman, 2004). If their support is bounded to $[0,1]$, they satisfy the same boundary constraints as warping functions. Therefore, we can exploit the vast literature on statistical distributions to define and characterize families of warping functions. Our family of warping functions is based on the two-sided power (TSP) distribution (Van Dorp & Kotz, 2002; Kotz & van Dorp, 2004).

### 3.1 BACKGROUND: TWO-SIDED POWER DISTRIBUTION

The TSP distribution has been proposed recently to model continuous random variables with bounded support $[a,b] \subset \mathbb{R}$. It generalizes the triangular distribution and can be viewed as a peaked alternative to the beta distribution (Johnson et al., 1994). In its most illustrative form, its probability density function (pdf) is unimodal with power-law decay on both sides, but it can yield U-shaped and J-shaped

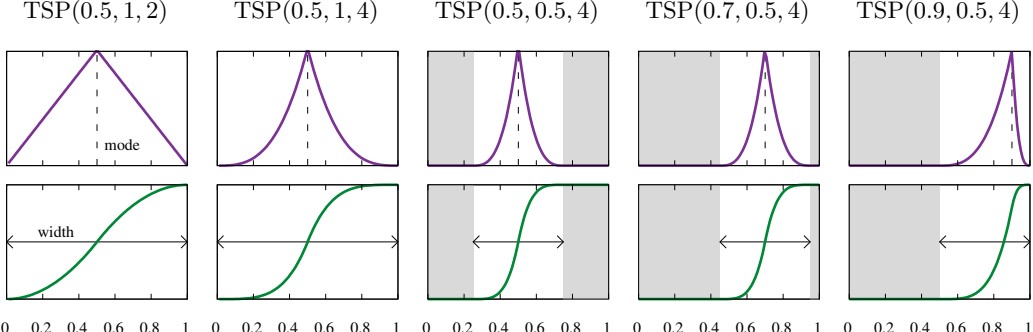

Figure 2: Three-parameter variant of the two-sided power distribution $\text{TSP}(m, w, n)$ on the interval $[0, 1]$. Dashed lines denote the modes $m$, arrows the widths $w$; shaded regions have probability zero. *Top row:* probability density function. *Bottom row:* cumulative distribution function.

pdfs as well, depending on the parametrization. Formally, the pdf is given by

$$f_{\text{TSP}}(u; a, m, b, n) = \begin{cases} \frac{n}{b-a} \left( \frac{u-a}{m-a} \right)^{n-1}, & \text{for } a < u \leq m \\ \frac{n}{b-a} \left( \frac{b-u}{b-m} \right)^{n-1}, & \text{for } m \leq u < b \\ 0, & \text{elsewhere,} \end{cases} \tag{8}$$

with $a \leq m \leq b$. $a$ and $b$ define the boundaries of the support, $m$ is the mode (anti-mode) of the distribution, and $n > 0$ is the power parameter that tapers the distribution. The triangular distribution is the special case with $n = 2$. In the following, we restrict our attention to the unimodal regime with $a < m < b$ and $n > 1$. In this case, the cdf is given by

$$F_{\text{TSP}}(u; a, m, b, n) = \begin{cases} 0, & \text{for } u \leq a \\ \frac{m-a}{b-a} \left( \frac{u-a}{m-a} \right)^n, & \text{for } a \leq u \leq m \\ 1 - \frac{b-m}{b-a} \left( \frac{b-u}{b-m} \right)^n, & \text{for } m \leq u \leq b \\ 1, & \text{for } b \leq u. \end{cases} \tag{9}$$

For convenience, we introduce a three-parameter variant of the TSP distribution with support restricted to *subintervals* of $[0, 1]$ located around the mode. It is fully specified by the mode $m \in (0, 1)$, the width $w \in (0, 1]$ of the subinterval, and the power $n > 1$. Depending on the mode and the width, the distribution is symmetric or asymmetric. Illustrations of the pdf and cdf of the three-parameter TSP distribution for various parametrizations can be found in Figure 2. We denote the three-parameter TSP distribution as $\text{TSP}(m, w, n)$ and write $f_{\text{TSP}}(u; m, w, n)$ and $F_{\text{TSP}}(u; m, w, n)$ for its pdf and cdf, respectively. The original parameters $a$ and $b$ are obtained from $m$ and $w$ via

$$a = \max \left( 0, \min \left( 1 - w, m - \frac{w}{2} \right) \right), \tag{10}$$

$$b = \min(1, a + w), \tag{11}$$

and yield a unimodal regime. Intuitively, the three-parameter TSP distribution describes a symmetric two-sided power kernel of window size $w$ that is located at $m$ and becomes asymmetric only if a symmetric window would exceed the domain $[0, 1]$. An advantage of the TSP distribution over the beta distribution is that its pdf and cdf have closed form expressions that are easy to evaluate computationally. Moreover, they are differentiable almost everywhere with respect to all parameters.

## 3.2 WARPING WITH MIXTURES OF TSP DISTRIBUTIONS

We define the TSP-based warping function $\gamma_{\text{TSP}} : [0, 1] \longrightarrow [0, 1]$ for a fixed number of segments $K$ as a *mixture distribution* of $K - 1$ three-parameter TSP distributions. Mixtures of unimodal distributions

Table 1: The relaxed segmented model with TSP-based warping functions.

| | | |
|---|---|---|
| data generating process | $x_t \mid z_t \overset{\text{iid}}{\sim} f_{\text{DGP}}\left(z_t, \hat{\theta}_t\right)$ | $t = 1, ..., T$ |
| parameter predictors | $\hat{\theta}_t := \sum_k \theta_k \max\left(0, 1 - \left|\hat{\zeta}_t - k\right|\right)$ | $t = 1, ..., T$ |
| segmentation predictors | $\hat{\zeta}_t := 1 + \gamma_{\text{TSP}}\left(\dfrac{t-1}{T-1}; m\right) \cdot (K-1)$ | $t = 1, ..., T$ |
| TSP mode predictors | $m_k := \dfrac{\sum_{k' \leq k} \exp(\mu_{k'})}{\sum_{k'=1}^{K} \exp(\mu_{k'})}$ | $k = 1, ..., K-1$ |
| segment parameters | $\theta_k$ | $k = 1, ..., K$ |
| TSP parameters | $\mu_k$ | $k = 1, ..., K$ |

have step-like cdfs that approximate segmentation functions. We use uniform mixture weights, and treat the width $w$ and power $n$ of the TSP component distributions as fixed hyperparameters. The components differ only in their modes $m = (m_1, ..., m_{K-1})$ with $m_k \in (0, 1)$:

$$\gamma_{\text{TSP}}(u; m) := \frac{1}{K-1} \sum_k F_{\text{TSP}}(u; m_k, w, n). \tag{12}$$

We constrain the modes to be strictly increasing, so that $\gamma_{\text{TSP}}$ is identifiable. If the windows around two consecutive modes $m_{k-1}$ and $m_k$ are non-overlapping, then $\gamma_{\text{TSP}}(u; m) = \frac{k-1}{K-1}$ between these windows. Furthermore, $\gamma_{\text{TSP}}(u; m) = 0$ before the first window and $\gamma_{\text{TSP}}(u; m) = 1$ after the last window. Therefore, the family of TSP-based warping functions contains piecewise-constant functions. The functions $\gamma_1$ and $\gamma_2$ in Figure 1 are examples of TSP-based warping functions. We show in Appendix A that any segmentation function can be represented by a TSP-based warping function.

## 4 MODEL ARCHITECTURE

We have described all components of the relaxed segmented model architecture. It can use any family of warping functions to approximate a segmentation function. An overview of the complete model architecture with *TSP-based warping functions* is provided in Table 1. To simplify the estimation problem, we rewrite the TSP modes as a normalized cumulative sum,

$$m_k := \frac{\sum_{k' \leq k} \exp(\mu_{k'})}{\sum_{k'=1}^{K} \exp(\mu_{k'})} \quad \text{for } k = 1, ..., K-1 \tag{13}$$

with unconstrained real parameters $\mu = (\mu_1, ..., \mu_K)$. The transformation of the parameters guarantees that the modes $m = (m_1, ..., m_{K-1})$ are strictly increasing and come from the interval $(0, 1)$. The warping function is now overparametrized, since the transformation is invariant to additive terms in the parameters $\mu$. This issue can be resolved by enforcing $\mu_1 := 0$.

The learnable parameters of this architecture are $\theta = (\theta_1, ..., \theta_K)$ for the DGP and $\mu = (\mu_1, ..., \mu_K)$ for the warping function. The hyperparameters are the number of segments $1 < K \ll T$, and the window size $w \in (0, 1]$ and power $n > 1$ of the TSP distributions. This architecture is a concatenation of simple functions that are either fully differentiable or differentiable almost everywhere. Therefore, all parameters can be learned jointly using gradient descent. A hard segmentation $\zeta$ of the input sequence can be obtained at any time during or after training by rounding $\hat{\zeta}_t$ to the nearest integers.

## 5 EXPERIMENTS

Our relaxed segmented model is highly versatile and can be employed for many different tasks. In Section 5.1, we study the performance of our approach in estimating a segmented generalized linear model (GLM) (Muggeo, 2003) on COVID-19 data. In Section 5.2, we evaluate our approach on

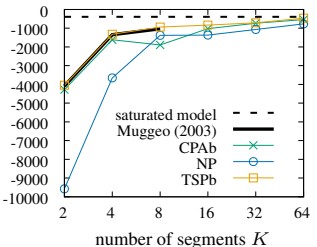 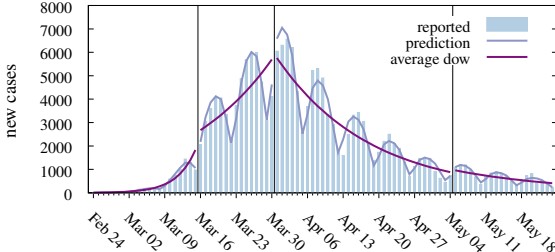

Figure 3: Segmented Poisson regression results on COVID-19 case numbers in Germany.

a change point detection benchmark against various competitors. In Section 5.3, we apply it on a streaming classification benchmark with concept drift. At last, Section 5.4 illustrates potential future applications of our model for discrete representation learning. We implemented our model in Python[1] using PyTorch[2] and optimize the parameters with ADAM (Kingma & Ba, 2015). We employ three different families of warping functions in our relaxed model: nonparametric (NP) (Lohit et al., 2019), CPA-based (CPAb) (Weber et al., 2019), and our TSP-based functions (TSPb). Source codes can be found in the supplementary material.

## 5.1 POISSON REGRESSION

Recent work has applied segmented Poisson regression to model COVID-19 case numbers (Küchen-hoff et al., 2020; Muggeo et al., 2020). We follow Küchenhoff et al. (2020) and model daily time series of *newly reported* COVID-19 cases during the first wave of the pandemic. We obtained official data for Germany from Robert Koch Institute.[3] A visualization of the reported data in Figure 3 (right plot, bars) reveals non-stationary growth rates and weekly periodicity. Therefore, we use time and a day-of-week indicator as covariates in the model. We tie the coefficients for the day-of-week indicators across all segments, while the daily growth rates and the bias terms differ in every segment. We estimate a standard segmented Poisson regression model with the baseline algorithm by Muggeo (2003), and our relaxed models (TSPb, CPAb, NP) with gradient descent. We estimate the baseline model and the relaxed models with $K = 2, 4, 8, 16, 32$, and $64$ segments. The true number of segments is unknown in this task. More details can be found in Appendix B.1.

Figure 3 (left plot) shows the goodness-of-fit (log-likelihood) of all models. TSPb consistently reaches the performance of the baseline algorithm. Moreover, TSPb consistently outperforms the other families of warping functions in this experiment. The improvement over NP is particularly large, which indicates that the "nonparametric" warping functions of Lohit et al. (2019) are harder to train than the parametric families. CPAb achieves a performance similar to TSPb, but the training time is much longer due to the more complex mathematical operations involved.

We observe that the goodness-of-fit generally grows with the number of segments $K$ and approaches the performance of a saturated model (one Poisson distribution per data point). For $K > 8$, the baseline of Muggeo (2003) terminates without a model estimate. Figure 3 (right plot) visualizes the best TSPb model for $K = 4$ (blue line). We also provide smoothed predictions where the average day of week (dow) effect is incorporated into the bias term to highlight the change of the growth rate from segment to segment (purple line). The three change points are located at 2020-03-16, 2020-03-31, and 2020-05-05. The baseline algorithm of Muggeo (2003) detects consistent change points at 2020-03-16 ($\pm0$ days), 2020-03-30 ($-1$ day), and 2020-05-01 ($-4$ days). Since the reported data is not iid within a segment (it is only conditionally iid given the covariates), other algorithms for change point detection cannot be applied as competitors on this task. Overall, the experiment shows that our model architecture allows effective training of segmented generalized linear models using gradient descent, in particular, when employed with TSPb warping functions.

---

[1]https://python.org/
[2]https://pytorch.org/
[3]https://www.rki.de/

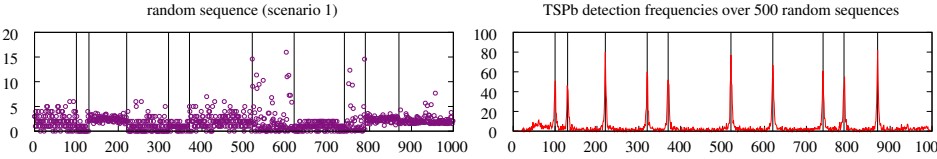

Figure 4: Change point detection task of Arlot et al. (2019) with detection results from our model.

Table 2: Empirical change point detection results.

| algorithm | sensitivity | #CPs (mean±std) | $d_{\text{hdf}}$ (mean±std) | $d_{\text{fro}}$ (mean±std) | reference |
|---|---|---|---|---|---|
| random | - | 10 | $127.8 \pm 45.5$ | $3.3 \pm 0.2$ | *baseline* |
| DP$\geq$1 | $\mu\sigma$ | 10 | $753.3 \pm 120.8$ | $4.1 \pm 0.3$ | Truong et al. (2020) |
| DP$\geq$10 | $\mu\sigma$ | 10 | $123.6 \pm 170.7$ | $\mathbf{2.1 \pm 0.7}$ | Truong et al. (2020) |
| BinSeg | $\mu\sigma$ | $10.0 \pm 3.1$ | $122.0 \pm 97.7$ | $2.5 \pm 0.4$ | Scott & Knott (1974) |
| PELT | $\mu\sigma$ | $86.2 \pm 26.7$ | $100.0 \pm 26.9$ | $8.7 \pm 1.5$ | Killick et al. (2012) |
| NP | $\mu\sigma$ | 10 | $88.4 \pm 35.7$ | $2.3 \pm 0.4$ | *this work* |
| CPAb | $\mu\sigma$ | 10 | $88.4 \pm 32.3$ | $2.7 \pm 0.3$ | *this work* |
| TSPb | $\mu\sigma$ | 10 | $\mathbf{82.5 \pm 32.3}$ | $2.3 \pm 0.3$ | *this work* |
| E-Divisive | * | $5.9 \pm 1.9$ | $162.0 \pm 108.2$ | $2.2 \pm 0.4$ | Matteson & James (2014) |
| KCP | * | $8.6 \pm 1.4$ | $67.3 \pm 55.1$ | $1.4 \pm 0.5$ | Arlot et al. (2019) |
| KCpE | * | 10 | $\mathbf{33.8 \pm 37.6}$ | $\mathbf{1.2 \pm 0.6}$ | Harchaoui & Cappé (2007) |

sensitivity: $\mu\sigma$ = mean/variance only, * = full distribution.
#CPs: number of change points; if equal to 10, #CPs is a parameter, otherwise, it is inferred automatically.

## 5.2 CHANGE POINT DETECTION

In the next experiment, we evaluate our relaxed segmented model on a change point detection task with simulated data. Our experimental design exactly follows Arlot et al. (2019). All details are given in Appendix B.2. We sample random sequences of length $T = 1000$ with 10 change points at predefined locations. For every segment in every sequence, a distribution is chosen randomly from a set of predefined distributions, and observations within that segment are sampled independently from that distribution. We follow scenario 1 of Arlot et al. (2019), where all predefined distributions have different means and/or variances. An example is shown in Figure 4 (left). We sample $N = 500$ such sequences, with change points at the same locations across all samples.

We apply our relaxed segmented model to estimate the change points. We model the data generating process by a normal distribution with different means and variances in every segment. This design choice makes our approach sensitive *only* to changes in the means and variances of the observed data, and no other distributional characteristics. We fit our model individually to all $N$ sequences to obtain individual estimates for the change points. Figure 4 (right) shows how many times a specific time step was detected as a change point by our approach (with TSPb warping functions). We observe clear peaks at the correct change point locations, which indicates that our model successfully recovers the original segmentations in many runs.

Table 2 summarizes the empirical detection performance of our approach (TSPb, CPAb and NP) and various competitors, including a baseline where change points are drawn randomly without replacement. It shows the Hausdorff distance $d_{\text{hdf}}$ and Frobenius distance $d_{\text{fro}}$ between the true segmentations and the detected segmentations (the lower the better). Among the approaches that detect changes in the mean and variance, our model performs best in terms of $d_{\text{hdf}}$ and second best in terms of $d_{\text{fro}}$, regardless of the choice of warping function. Note that all approaches in this group optimize the same objective function (the likelihood of the data under a normal distribution). The dynamic programming approaches (DP$\geq\ell$) (Truong et al., 2020) exactly find the optimal solution for a predefined number of change points, with a minimum segment length of $\ell$. PELT (Killick et al., 2012) finds the optimal solution with an arbitrary number of change points, while BinSeg (Scott & Knott, 1974) approximates that solution. Although our gradient-based method finds suboptimal solutions in terms of the normal likelihood, it produces results with the lowest segmentation costs. This indicates a regularizing effect of the relaxed segmented model that avoids degenerate segmentations. Our

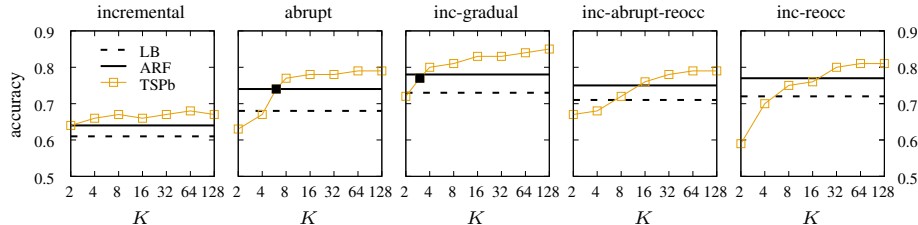

Figure 5: Classification performance on the insect stream benchmark of Souza et al. (2020).

approach is only outperformed by algorithms for kernel-based change point detection (Harchaoui & Cappé, 2007; Arlot et al., 2019) that are sensitive towards all distributional characteristics.

## 5.3 CONCEPT DRIFT

A key novelty of our segmented model architecture is that it allows joint training of the segmentation *and any other model component* using gradient descent. As a proof of concept, we apply our model on a classification problem with concept drift (Gama et al., 2013). The model has to learn the points in time when the target concepts change, and a useful feature transformation for the task. We use the insect stream benchmark of Souza et al. (2020) for this purpose. The task is to classify insects into 6 different species using 33 features collected from an optical sensor. The challenge is that these species behave differently when the air temperature (which is not included as a feature) changes. The benchmark contains multiple data streams, where the air temperature is controlled in different ways (incremental, abrupt, incremental-gradual, incremental-abrupt-reoccurring, incremental-reoccurring). The classifier must adapt the learned concepts depending on the current air temperature.

We employ our relaxed segmented model for softmax regression with the cross entropy loss to obtain segmentations of the data streams. We focus on the five data streams with balanced classes from the benchmark, and measure performance by the classification accuracy. We fit models with $K = 2, 4, 8, ..., 128$ segments using TSPb warping functions. We transform the input instances with a fully connected layer followed by a ReLU nonlinearity before passing them to the segmented model. The feature transformation is shared across all segments. We jointly learn the parameters of the segmented model and the feature transformation with gradient descent. Details can be found in Appendix B.3. Results are visualized in Figure 5. In addition to our own results, we include the prequential accuracies of the strongest competitors reported by Souza et al. (2020): Leveraging Bagging (LB) (Bifet et al., 2010) and Adaptive Random Forests (ARF) (Gomes et al., 2017).

Our relaxed segmented model reaches and outperforms the strongest competitors on all streams from the benchmark, if the number of segments $K$ is large enough to accommodate the concept drift present in the stream. Note that only the data stream with *abrupt* concept drift satisfies our modeling assumption of a piecewise stationary data generating process. It consists of six segments with constant air temperatures within each segment. The *incremental-gradual* stream has three segments, with mildly varying temperatures in the outer segments and mixed temperatures in the inner segment. The black boxes in their plots show the performances achieved with our approach for $K = 6$ and $K = 3$ segments, respectively. The results indicate that these are in fact the minimum numbers of segments required to obtain competitive performance on these data streams.

## 5.4 DISCRETE REPRESENTATION LEARNING

At last, we apply our model on a speech signal to showcase its potential for discrete representation learning on the level of phonemes. We assume that the speech signal—represented by a sequence of 12-dimensional MFCC vectors—is piecewise constant within a phoneme. We model it by a minimal DGP with no covariates that simply copies the parameter vectors to the output. See Appendix B.4 for a complete model description. We fit the model to a single utterance ("choreographer", 10 phonemes) from the TIMIT corpus (Garofolo et al., 1993) by minimizing the mean squared error loss, and obtain:

Although the simple DGP does not capture all dynamics of the speech signal, 7 out of 9 phoneme boundaries were correctly identified, with a time tolerance of 20 ms. This minimal experiment

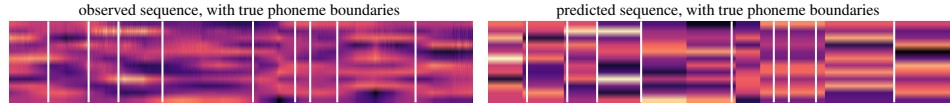

suggests that relaxed segmented models, when combined with more powerful DGPs, may be useful for discrete representation learning (Rolfe, 2017; van den Oord et al., 2017; Fortuin et al., 2019), in particular for learning segmental embeddings (Kong et al., 2016; Wang et al., 2018; Chorowski et al., 2019; Kreuk et al., 2020). In fact, our relaxed segmented model may be part of a larger model architecture, where the covariates $z_t$ and the parameters $\theta_k$ come from some upstream computational layer, and the outputs $x_t$ are passed on to the next computational layer with an arbitrary downstream loss function. The interpretation of $z_t$ as *covariates* and $\theta_k$ as *parameters* is merely for consistency with prior work on segmented models. It is more accurate to interpret $z_t$ as *temporal variables* that differ for every time step $t$, and $\theta_k$ as *segmental variables* that differ for every segment $k$. The DGP combines the information from both types of variables to produce an output for every time step.

## 6 CONCLUSION

We have described a novel approach to learn segmented models for non-stationary sequential data with discrete change points. Our relaxed segmented model formulation is highly versatile and can use any family of warping functions to approximate a hard segmentation function. If the family of warping functions is differentiable, our model can be trained with gradient descent. A key advantage of a differentiable model formulation is that it enables the integration of state-of-the-art learning architectures within segmented models. We have introduced the novel family of TSP-based warping functions designed specifically for the segmentation task: it is differentiable, contains piecewise-constant functions that exactly represent segmentation functions, its parameters directly correspond to segment boundaries, and it is simple to evaluate computationally. The experiments on diverse tasks demonstrate the modeling capacities of our approach.

**Limitations and future work.** The most immediate limitation of our relaxed segmented model that it shares with classical segmented models (Bai & Perron, 2003; Muggeo, 2003) is that the number of segments $K$ is a hyperparameter and needs to be chosen prior to model fitting. This issue can be resolved *externally* with any model selection criterion (Davis et al., 2006). However, due to our differentiable model formulation, future work may be able to solve the model selection task *internally* within a single objective function that is optimized with gradient descent, e.g., using the differentiable architecture search approach of Liu et al. (2019).

Another limitation that we inherit from classical segmented models is that all $K$ segments are treated as distinct regimes with their own parameter vectors $\theta_k$. For example, if the data-generating process switches back and forth multiple times between only two distinct regimes, a segmented model has to learn the same parameter vector every time the regime switches. During training, we can favor solutions with reused segment parameters by adding a regularization term to the loss function that penalizes a high rank of the segment parameter matrix $\theta = [\theta_1, ..., \theta_K]$. If we wanted to control the number of distinct regimes separately, we could learn a codebook of $C < K$ distinct segment parameters and a mapping from segment identifiers to codebook entries $f : \{1, ..., K\} \rightarrow \{1, ..., C\}$, e.g., using the vector quantization approach of van den Oord et al. (2017). This is possible only because our model is fully differentiable and can easily be augmented with such components.

At last, we consider the application of our approach for discrete representation learning under powerful deep data-generating processes sketched in Section 5.4 a fruitful direction for future work.

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

## A    WARPING FUNCTIONS FOR SEGMENTATION

In this section, we provide a technical notion of representation of segmentation functions and show that TSP-based warping functions can represent segmentation functions in this sense.

**Definition 1.** The warping function $\gamma : [0, 1] \longrightarrow [0, 1]$ **exactly represents** the segmentation function $\zeta : \{1, ..., T\} \longrightarrow \{1, ..., K\}$ with respect to the unit grid $(u_t)_{t=1}^{T}$ if

$$\gamma(u_t) = \frac{k-1}{K-1} \Leftrightarrow \zeta(t) = k \quad \text{for all } t. \tag{14}$$

Exact representation entails that the predictors $\hat{\zeta}_t$ defined in Equation 6 satisfy $\hat{\zeta}_t = \zeta(t)$ for all $t$. Clearly, exact representation can only be achieved with piecewise-constant warping functions. We have the following result:

**Lemma 1.** *For every segmentation function $\zeta$, there is a TSP-based warping function $\gamma_{TSP}$ that exactly represents $\zeta$.*

*Proof.* We place the $K - 1$ modes $m_k$ of $\gamma_{\text{TSP}}$ on the change points $\tau_k$ (projected to the unit grid) and choose a window size $w$ not larger than the resolution of the grid. The power $n > 1$ can be chosen freely. Formally, let $\tau_k \in \{2, ..., T\}$ be the $k$-th change point, such that $\zeta(\tau_k - 1) = k$ and $\zeta(\tau_k) = k + 1$. We set $m_k := (u_{\tau_k} + u_{\tau_k - 1})/2$ for all $k = 1, ..., K - 1$, and $w := 1/(T - 1)$. $\quad\square$

In practice, the segmentation function $\zeta$ is unknown and the modes $m = (m_1, ..., m_{K-1})$ must be estimated in an unsupervised way. For effective training with gradient descent, the window size should initially be *larger* than the sampling resolution of the unit grid, $w > 1/(T - 1)$, to allow the loss to backpropagate across segment boundaries. The window size can be interpreted as the *receptive field* of the individual TSP components. The width can be tapered down to $w \leq 1/(T - 1)$ over the training epochs to obtain a warping function that exactly represents a segmentation function. For simplicity, we do not follow this strategy in our experiments, but instead round the predictors $\hat{\zeta}_t$ to the nearest integers, whenever we need hard segmentations.

## B    DETAILS ON THE EXPERIMENTS

### B.1    POISSON REGRESSION

**Data.**    We obtained official data on COVID-19 cases in Germany from Robert Koch Institute (RKI), the German public health institute. The data is publicly available under an open data license.[4] We provide a copy used for the experiments in this work in the supplementary material. For every day in the study period, we aggregate all new cases *reported* on that day. Due to the delays between the actual infection of a patient and the time the infection is reported to the health authorities, the new cases reported on a specific day contain new infections from several days before.

**Segmented Poisson regression model.**    Poisson regression is a GLM for count data, where the data generating process is modeled by a Poisson distribution and the linear predictor is transformed with the logarithmic link function (McCullagh & Nelder, 1989).

Let $x_t$ denote the number of newly reported cases at time $t$. Furthermore, let $z_t^{\text{Tu}}, z_t^{\text{We}}, z_t^{\text{Th}}, z_t^{\text{Fr}}, z_t^{\text{Sa}}$ and $z_t^{\text{Su}}$ denote binary day-of-week indicators. If the time step $t$ is a Monday, all indicators are 0. For all other days, exactly one indicator is set to 1. We use the following vector of covariates, including a bogus covariate 1 for the bias terms:

$$z_t = [1, t, z_t^{\text{Tu}}, z_t^{\text{We}}, z_t^{\text{Th}}, z_t^{\text{Fr}}, z_t^{\text{Sa}}, z_t^{\text{Su}}] \tag{15}$$

In our segmented Poisson regression model, the bias terms and the daily growth rates (the parameters associated with the covariates 1 and $t$) differ in every segment, while the parameters for the day-of-week indicators are tied across all segments (they are independent of segment $k$):

$$\theta_k = [\theta_{k,1}, \theta_{k,2}, \theta^{\text{Tu}}, \theta^{\text{We}}, \theta^{\text{Th}}, \theta^{\text{Fr}}, \theta^{\text{Sa}}, \theta^{\text{Su}}] \tag{16}$$

---

[4]"Fallzahlen in Deutschland" by Robert Koch Institut (RKI), open data license "Data licence Germany – attribution – Version 2.0" (dl-de/by-2-0), link: https://www.arcgis.com/home/item.html?id=f10774f1c63e40168479a1feb6c7ca74

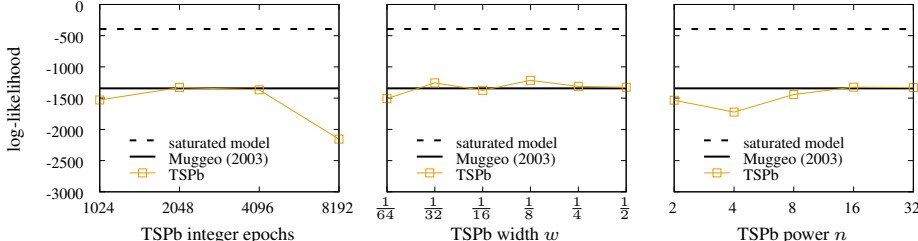

Figure 6: Ablation study for TSPb hyperparameters (segmented Poisson regression); default values for the fixed hyperparameters are: 2048 integer epochs, $w = .5$, and $n = 16$, respectively.

In this model, the bias terms $\theta_{k,1}$ control the base rates of newly reported cases *on a Monday* within every segment $k$, while the parameters for the day-of-week indicators are global scaling factors that control the relative increase or decrease of the base rates on the other days-of-week *with respect to Monday*. Overall, we have the following segmented Poisson regression model:

$$\hat{x}_t = \mathbb{E}[x_t \mid z_t] := \exp\left(\theta_{k,1} + \theta_{k,2} \cdot t + \sum_{\text{day} \in \{\text{Tu,We,Th,Fr,Sa,Su}\}} \theta^{\text{day}} z_t^{\text{day}}\right), \text{ if } \zeta(t) = k \qquad (17)$$

The training objective is to minimize the negative log-likelihood under a Poisson distribution:

$$\mathcal{L}(\zeta, \theta) = -\sum_t \log \text{Poisson}(x_t \mid \hat{x}_t) \qquad (18)$$

In our TSP-based warping functions (TSPb), we set the window size to $w = .5$ and the power to $n = 16$. In the CPA-based warping functions (CPAb) of Weber et al. (2019), we set the dimensionality of the underlying velocity fields to $K$, the number of segments to learn. This choice makes the function family flexible enough to approximate warping functions with $K - 1$ discrete steps, using the minimum number of parameters necessary. Note that the "nonparametric" warping functions (NP) of Lohit et al. (2019) have $T - 1$ parameters, where $T$ is the sequence length. We perform training with ADAM with a learning rate of $\eta = 0.01$ for a total of 10000 training epochs. In the last 2048 epochs, we round the predictors $\hat{\zeta}_t$ to the nearest integers to obtain a hard segmentation function $\zeta(t)$. We perform 10 restarts of the training procedure with random initialization and keep the model with the best fit for evaluation (highest log-likelihood).

**Competitor.** We use the reference implementation of Muggeo (2003) from the R `segmented` package. As pointed out in Section 1, our segmented model architecture learns a segmentation of the *index set* $t = 1, ..., T$. The segmented models by Muggeo (2003) learn a segmentation in the *domain of one (or more) of the covariates*. Since we use the index $t$ as a covariate, we can configure the algorithm of Muggeo (2003) to segment the covariate $t$, which makes the two models equivalent.

**Ablation study.** Figure 6 shows the goodness-of-fit (log-likelihood, the higher the better) obtained with our relaxed segmented model (TSPb) using different settings of the hyperparameters. The performance is quite robust with respect to the width and power of the TSP components. The number of integer epochs (with hard segmentations) has impact on the estimation performance. We hypothesize that the learning problem is simpler with soft segmentations, so that gradient descent moves towards a better region of the loss function. During integer epochs, the parameters within a segment are fine-tuned within the region found during the soft epochs.

### B.2 CHANGE POINT DETECTION

**Data generating process.** We follow scenario 1 of Arlot et al. (2019) to sample $N = 500$ random sequences of length $T = 1000$ with a total of 10 change points ($K = 11$ segments). The change points are located at time steps 100, 130, 220, 320, 370, 520, 620, 740, 790, and 870. We define a change point as the *beginning* of a new segment. The data generating process is described in

---

**Algorithm 1** Data generating process of Arlot et al. (2019).

---

**for** all sequences $n = 1, ..., N$ **do**
    **for** all segments $k = 1, ..., K$ **do**
        sample a distribution $p_{nk}$ uniformly from $\mathcal{P} \setminus \{p_{n,k-1}\}$
        sample $x_{nt} \stackrel{\text{iid}}{\sim} p_{nk}$ for all time steps $t$ in segment $k$
    **end for**
**end for**

---

Algorithm 1, where $\mathcal{P}$ is a set of predefined probability distributions. For scenario 1, we have

$$\mathcal{P} := \{ \text{Binom}(n = 10, p = 0.2), \text{NegBinom}(n = 3, p = 0.7),$$
$$\text{Hypergeom}(M = 10, n = 5, N = 2), \mathcal{N}(\mu = 2.5, \sigma^2 = 0.25), \quad (19)$$
$$\Gamma(a = 0.5, b = 5), \text{Weibull}(a = 2, b = 5), \text{Pareto}(a = 3, b = 1.5)\},$$

where $a$ is the shape parameter and $b$ is the scale parameter in the case of $\Gamma$, Weibull, and Pareto. For every algorithm in the evaluation, we create a new sample of 500 sequences.

**Segmented normal model.** We employ our relaxed segmented model with the assumption that the data generating process within each segment is a *normal distribution* with its own mean and variance,

$$x_t \stackrel{\text{iid}}{\sim} \mathcal{N}(\mu_k, \sigma_k^2), \quad \text{if } \zeta(t) = k. \quad (20)$$

With this design choice, we can detect changes in the mean and variance between the segments, but no other distributional characteristics. The training objective is to minimize the negative log-likelihood:

$$\mathcal{L}(\zeta, \theta) = -\sum_t \log \mathcal{N}(x_t \mid \mu_{\zeta(t)}, \sigma_{\zeta(t)}^2) \quad (21)$$

We ensure a positive variance throughout training by estimating the logarithm of the variance, i.e., the parameter vector within a segment $k$ is given by $\theta_k = [\mu_k, \log(\sigma_k^2)]$.

For TSPb, we set the window size to $w = .125$ and the power to $n = 16$. For CPAb, we set the dimensionality of the velocity field to $K$. NP has no hyperparameters. We perform training with ADAM with a learning rate of $\eta = 0.1$ for a total of 300 epochs with 100 integer epochs. We perform 10 restarts with random initialization and keep the model with the best fit for evaluation.

**Competitors.** For the experiments with the dynamic programming approaches (DP $\geq \ell$), we use the implementation of Truong et al. (2020) from the Python `ruptures` module, using the normal cost function with a minimum segment size of $\ell$, 10 change points, and no subsampling. For the binary segmentation approach (BinSeg) by Scott & Knott (1974) and the Pruned Exact Linear Time (PELT) method of Killick et al. (2012), we use the R `changepoint` package (`cpt.meanvar`, normal test statistic, MBIC penalty (Zhang & Siegmund, 2007), minimum segment size 2, maximum number of change points 100). The experiments with the E-divisive approach (Matteson & James, 2014) were conducted with the R `ecp` package (`e.divisive`, significance level .05, 199 random permutations, minimum, minimum segment size 2, moment index 1). For the experiments with the kernel-based approaches KCP (Arlot et al., 2019) and KCpE (Harchaoui & Cappé, 2007), we use the implementation from the Python `chapydette` package kindly provided by Jones & Harchaoui (2020), with the default Gaussian-Euclidean kernel, bandwidth 0.1, and minimum segment size 2.

**Performance measures.** We use the same evaluation measures as Arlot et al. (2019) and follow their definitions. Let $\zeta$ and $\zeta'$ be two segmentations of a sequence of length $T$. Let $\tau$ and $\tau'$ be the corresponding sets of change points. The Hausdorff distance is the largest distance between any change point from one segmentation and its nearest neighbor from the other segmentation:

$$d_{\text{hdf}}(\zeta, \zeta') := \max \left\{ \max_{1 \leq i \leq |\tau|} \min_{1 \leq j \leq |\tau'|} |\tau_i - \tau_j'|, \max_{1 \leq j \leq |\tau|} \min_{1 \leq i \leq |\tau'|} |\tau_i - \tau_j'|, \right\} \quad (22)$$

Table 3: Ablation study for TSPb hyperparameters (change detection).

| power $n$ | width $w$ | $d_{\text{hdf}}$ (mean±std) | $d_{\text{fro}}$ (mean±std) |
|---|---|---|---|
| 16 | 0.5 | $78.8 \pm 31.9$ | $2.4 \pm 0.3$ |
| 16 | 0.25 | $83.3 \pm 35.1$ | $2.3 \pm 0.3$ |
| 16 | 0.125 | $82.5 \pm 32.3$ | $2.3 \pm 0.3$ |
| 16 | 0.0625 | $85.0 \pm 36.1$ | $2.4 \pm 0.3$ |
| 4 | 0.125 | $83.0 \pm 34.4$ | $2.4 \pm 0.3$ |
| 8 | 0.125 | $79.5 \pm 31.3$ | $2.3 \pm 0.4$ |
| 16 | 0.125 | $82.5 \pm 32.3$ | $2.3 \pm 0.3$ |
| 32 | 0.125 | $80.4 \pm 30.1$ | $2.3 \pm 0.3$ |

The Frobenius distance between two segmentations (Lajugie et al., 2014) is defined as the Frobenius distance between the *rescaled equivalence matrix* representations of the segmentations:

$$d_{\text{fro}}(\zeta, \zeta') := \|M^\zeta - M^{\zeta'}\|_F, \tag{23}$$

$$\text{where} \quad M^\zeta_{t,t'} = \frac{\mathbf{1}_{\zeta(t)=\zeta(t')}}{\sum_{t''} \mathbf{1}_{\zeta(t)=\zeta(t'')}}. \tag{24}$$

It penalizes over-segmentation more strongly than the Hausdorff distance.

**Ablation study.** We experimented with different values for the TSPb hyperparameters; Table 3 shows the results. The detection performance is robust towards the choice of hyperparameters.

### B.3 CONCEPT DRIFT

**Benchmark data.** The insect stream benchmark is described in detail in Souza et al. (2020). The five data streams with balanced class distributions that we consider here have the following lengths. *incremental*: 57,018; *abrupt*: 52,848; *incremental-gradual*: 24,150; *incremental-abrupt-reoccurring*: 79,986; *incremental-reoccurring*: 79,986 (sic).

**Softmax regression model.** Softmax regression is a standard multi-class classification model, where the data generating process is modeled by a categorical distribution with probabilities computed from the softmax function. Let $(x_t, y_t)$ denote a training instance from the insect stream benchmark, where $x_t \in \{1, ..., C\}$ is the target class label and $y_t$ is the raw observation. We learn a more informative representation $z_t$ of the observation $y_t$ by passing it through a linear layer with output dimension $D = 8$, followed by a ReLU nonlinearity. We denote this feature transformation by $z_t := f_\phi(y_t)$, where $\phi$ are the learnable parameters of the transformation. The covariates $z_t$ are used within a segmented softmax regression model to predict the targets $x_t$,

$$x_t \mid z_t \overset{\text{iid}}{\sim} \text{Categorical}(\text{Softmax}(\theta_k z_t)), \quad \text{if } \zeta(t) = k. \tag{25}$$

In this model, $\theta_k \in \mathbb{R}^{C \times D}$ is a matrix, so that $\theta_k z_t \in \mathbb{R}^C$ contains unnormalized classification scores for every class $c$ in segment $k$. The softmax function transforms the scores into normalized class probabilities. The feature transformation $f_\phi$ is shared across all segments, while the parameters of the linear predictor $\theta$ change from segment to segment. With this design, we learn a feature transformation that is useful for the classification task in general, while taking concept drift in the label associations into account. The training objective is to minimize the negative log-likelihood under the categorical distribution, more commonly known as the cross-entropy loss

$$\mathcal{L}(\zeta, \theta, \phi) = -\sum_t \log \text{Categorical}(x_t \mid \text{Softmax}(\theta_{\zeta(t)} z_t)) \tag{26}$$

$$= -\sum_t \left( \left[\theta_{\zeta(t)} z_t\right]_{x_t} - \log \sum_c \exp \left[\theta_{\zeta(t)} z_t\right]_c \right). \tag{27}$$

We employ TSPb warping functions with window size $w = .125$ and power $n = 16$. We perform training with ADAM with a learning rate of $\eta = .1$ for a total of 300 epochs, with 100 integer epochs. We perform 10 restarts with random initialization and keep the model with the best fit for evaluation.

### B.4 DISCRETE REPRESENTATION LEARNING

**Piecewise constant model.** We assume the piecewise constant, deterministic DGP

$$x_t := \theta_k, \quad \text{if } \zeta(t) = k, \tag{28}$$

and minimize the mean squared error of the output

$$\mathcal{L}(\zeta, \theta) = \sum_t \|x_t - \theta_k\|^2. \tag{29}$$

We fit our relaxed segmented model with TSPb warping functions ($K = 10$ segments) with window size $w = .125$ and power $n = 16$. We perform training with ADAM with a learning rate of $\eta = .1$ for a total of 300 epochs with 100 integer epochs. We perform 10 restarts with random initialization and keep the model with the best fit for evaluation.

