# OpenReview forum: "Differentiable Segmentation of Sequences"
_ICLR.cc/2021/Conference — ICLR 2021 Poster_

### Official Review · AnonReviewer4 · 2020-10-28
**Elegant differential segmentation representation; Limited evaluation**

**Rating:** 6
**Confidence:** 3

**Review:**

The paper describes the use of two-sided power functions for differentiable approximation of segmentation in discrete sequences with monotonic segmentation (each event in the sequence is defined by one continuous interval). The authors show that their particular parametrization allows to control exactly the length, starting, and ending point of the interval, and the slope of change (Fig. 2).
The main result of the paper is the ability to pose segmentation and segment estimation problem jointly and differentiably without assumptions on the model that produced the segment data, which is an improvement over several recent works with could also pose the problem jointly, but made such assumptions, such as van den Burg & Williams, 2020, and Arlot et al., 2019.

The authors show results on several tasks: modelling of COVID-19, change point detection on synthetic dataset from Arlot et al, 2019, and concept drift dataset. The results on change point detection and concept drift dataset are convincing, showing better quality of segmentation than previous approaches. The results on COVID-19 are, though, rather qualitative and serve more as an ablation study as there aren't many different results on this datasets.

My biggest issue is the absence of proper evaluation on harder tasks, where the estimation problem is solved with the help of a neural network, as opposed to something else, as in the previous 3 datasets.
Authors show results on speech segmentation without any comparison to other approaches except for random segmentation (with a comment that "random segmentation is worse in 9932 out of 10000 trials (99.32%)") in the "eval-timit colab", which is hardly a fair comparison). There is a number of baseline techniques and datasets for segmentation that the authors could compare to. For example, one way of extracting segmentation in OCR / Speech is by looking at logits for "empty class" coming from the decoder and it would be interesting to compare such segmentation with a task of segmentation + character recognition formulated in this framework. Another group of approaches, identified by the authors themselves, uses discrete latent space representation for segmentation, in the domain of Speech. Finally, in the domain of action segmentation  there are similar in nature approaches also predicting a differentiable segmentation, but without explicit parametrization, such as in this work (eg. [Fast Weakly Supervised Action Segmentation Using Mutual Consistency])

One other question is regarding section 5.3 and Fig. 6. It seems that increasing the number of segments up to 32 still produces improvements in accuracy, so I would be interested in seeing how the model behaves with even larger K. (I expect that starting at some value of K overfitting to small segments should start reducing the accuracy?)

I believe paper could be strengthened by evaluation on harder tasks and comparison with other types of methods. That being said, I find the described approach valuable in and of itself, hence the rating.

---

> ### Author Response · Authors · 2020-11-13
> **Please mind the differences between the tasks**
>
> Thank you for the critical remarks!
>
> > My biggest issue is the absence of proper evaluation on harder tasks, where the estimation problem is solved with the help of a neural network, as opposed to something else, as in the previous 3 datasets.
>
> We believe that our main contribution lies in the differentiable formulation of the learning problem for segmented models, and we try to show with our evaluation that this formulation actually works and reaches/outperforms competitors on three very different tasks. In the third task (concept drift), we actually employ a neural network, although, admittedly, a shallow one (see Section B.3).
>
> That being said, we are highly interested in evaluating our approach (or variations thereof) also on harder tasks that require deep architectures. For this reason, in Section 5.4, we highlight one possible future application of our approach for discrete representation learning in the context of speech segmentation. We do not claim that the simple model presented there actually solves the speech segmentation problem, which is why we did not perform a formal evaluation against competitors from the speech recognition community.
>
> > There is a number of baseline techniques and datasets for segmentation that the authors could compare to. [...] OCR / Speech [...] action segmentation [...]
>
> It is true that these problems are similar in nature, since they all require some form of segmentation. However, technically, they are quite different from the problem we consider. In the present work, we consider the problem of segmenting a single sequence of length $T$ into a fixed number of segments $K$ under a generic model for the data generating process (which is a hard problem). The problems you mention require segmentation of multiple sequences of different lengths into an unknown number of segments. In order to apply our segmented model to these problems we would have to come up with non-trivial extensions that would exceed the scope of the present work.
>
> Another perspective on the difference is the following: With our approach, we learn the parameters of a segmentation function as a part of fitting a segmented model. In the problems you mention, a model is trained to predict a segmentation for an arbitrary input sequence. We believe that these problems can be unified in future work.
>
> > One other question is regarding section 5.3 and Fig. 6. It seems that increasing the number of segments up to 32 still produces improvements in accuracy, so I would be interested in seeing how the model behaves with even larger K. (I expect that starting at some value of K overfitting to small segments should start reducing the accuracy?)
>
> Yes, the model will eventually start to overfit the data. However, we note that this is a streaming classification benchmark without a train/test split, and we fit our segmented model to the complete stream. In the case of extreme overfitting with $K=T$ (a saturated model), the segmented model will have a classification accuracy of 1, since it can adapt the bias term individually for every instance such that it is correctly classified. Therefore, we will not see the accuracy going down with $K \longrightarrow T$. We still ran additional experiments with up to 128 segments and added the results to Figure 5 in the updated manuscript.

---

### Official Review · AnonReviewer2 · 2020-10-31
**Official Blind Review #2**

**Rating:** 7
**Confidence:** 3

**Review:**

[summary]
The paper proposes a relaxed way to solve the segmentation of sequence that can directly leverage the deep learning architectures. The relaxed model allows each segmentation parameter to be a linear interpolations between two consecutive parameters depends on a continuous warping function. The paper then proposes to use mixture of TSP distribution to simulate a step-like warping function and perform a thorough empirical comparison results with different methods and different warping functions. It turns out that TSP-based methods consistently achieve good performance.

[novelty]
To be honest I am not familiar with this area. It seems that this is a very novel method and may have impact across the area.

[significance]
The algorithm is very easy to understand and simple to implement, which may be very easy to reproduce by the community. The segmentation problem itself, especially the COVID19 case, show the importance of this area and may attract further attention from even outside the community.

[clarity]
I enjoy reading the paper and find it very easy to follow. The experimental results are clear and detailed.

[some further questions]
I'm curious for the relaxed models in equation (4). I didn't see why (4) should be very general form of relaxation. Why is it important to only interpolate consecutive two parameters? Is is possible to rewrite (4) into a weighted average $\sum_k w_{k,t} \theta_k$ and we hope so that $w_{k,t}$ depend on a continuous function, similar to $\zeta_t$, and index $k$?

---

> ### Author Response · Authors · 2020-11-11
> **Interpolation enforces monotonicity**
>
> Thank you for the positive feedback!
>
> > I'm curious for the relaxed models in equation (4). I didn't see why (4) should be very general form of relaxation. Why is it important to only interpolate consecutive two parameters? Is is possible to rewrite (4) into a weighted average $\sum_k w_{k,t}\theta_k$ and we hope so that $w_{k,t}$ depend on a continuous function, similar to $\zeta_t$, and index $k$?
>
> Yes, we could provide more general forms of relaxations. The weights $w_{k,t}$ that you mention implicitly yield a stochastic alignment matrix $W = (w_{k,t})_{k,t}$ that aligns time steps to segments. In the most general formulation of the model, the only constraint on the matrix $W$ is that $\sum_k w_{k,t} = 1$ for all $t$. For example, with $K=4$ and a sequence of length $T=30$, this approach could yield the segmentation
>
> 134422123341123344422123343324
>
> with a large number of "effective" segments due to the lack of temporal smoothness/monotonicity in the alignment matrix. This may be useful for some applications, but requires learning a (more or less) unconstrained $K \times T$ matrix. We want the alignment matrix to fulfil a monotonicity constraint: the segments should be visited monotonically from segment $1$ to segment $K$. The way we obtain the weights (by interpolating a monotonic warping function) enforces this monotonicity, such that the segmentation has the form
>
> 111111111222222333333333444444
>
> When using TSP-based warping functions, the (implicit) $K \times T$ alignment matrix is effectively parametrized with only $K$ parameters. We could replace the linear interpolation in Equation (4) by quadratic, cubic, or higher-order interpolation and still obtain a monotonic segmentation. However, we chose linear interpolation because it is the simplest variant, worked well, and was used to learn differentiable warping functions before, e.g., in refs [1,2,3].
>
> We made this aspect of the model more explicit in the updated manuscript.
>
> [1] Nicki Skafte Detlefsen, Oren Freifeld, and Soren Hauberg. Deep Diffeomorphic Transformer Networks. In: CVPR, 2018.
>
> [2] Ron Shapira Weber, Matan Eyal, Nicki Skafte Detlefsen, Oren Shriki, and Oren Freifeld. Diffeomorphic Temporal Alignment Nets. In: NeurIPS, 2019.
>
> [3] Suhas Lohit, Qiao Wang, and Pavan Turaga. Temporal Transformer Networks: Joint Learning of Invariant and Discriminative Time Warping. In: CVPR, 2019.

---

### Official Review · AnonReviewer1 · 2020-11-06
**Novel method, interesting experiments, easy to read**

**Rating:** 7
**Confidence:** 3

**Review:**

The proposed paper introduces a novel approach for the segmentation of sequences. The proposed method is based on two-sided power distributions (TSP) that are mathematically well-define and enable differentiability. The main goal of the method is to jointly optimize model parameters, including the segmentation function. The method is validated through experiments on modeling the spread of COVID-19 based on Poisson regression, change point detection, a classification model with concept drift (insect stream benchmark), and discrete representation learning (speech signal).

Overall, I have the impression that this is in interesting paper that has most things going for it. Replacing a hard segmentation function with a soft differentiable warping function (two-sided power distributions) seems technically-sound and is an interesting novel solution to a difficult problem. The paper is mostly well-written and easy to follow. Furthermore, the experiments are well-defined and novel datasets (COVID, insect stream benchmark) were used to validate the effectiveness of the model. Therefore, I am leaning toward accepting this work to ICLR 2021.

Additional comments:

- The abstract is somewhat difficult to comprehend and appears more cryptic than necessary.
- In Section 3.2. it is not clear what is meant by levels 0 and 1.
- Section 4: It is not clear what 'bogus mode' refers to.
- To better understand the model in Figure 3 (and given the white space around the equations) it would be helpful to provide  labels for the introduced variables.
- Please add details on whether the RKI data is publicly available.
- Currently the paper does not discuss any limitations. To further understand the introduced model it would be helpful to highlight corner cases across the experiments in which the model does not perform well.

---

> ### Author Response · Authors · 2020-11-12
> **Some clarifications, main limitation**
>
> Thank you for pointing us to aspects that remained unclear!
>
> > In Section 3.2. it is not clear what is meant by levels 0 and 1.
>
> > Section 4: It is not clear what 'bogus mode' refers to.
>
> We tried to clarify these points by removing the informal terms 'level' and 'bogus mode' in the updated manuscript and expressing the statements mathematically. Please let us know if further changes are necessary.
>
> > To better understand the model in Figure 3 (and given the white space around the equations) it would be helpful to provide labels for the introduced variables.
>
> Thank you for the hint! We replaced Figure 3 with the new Table 1 that provides a more comprehensive overview of the model architecture.
>
> > Please add details on whether the RKI data is publicly available.
>
> Yes, it is publicly available and in fact, open data. The Jupyter notebook eval-covid19 in the supplementary material has detailed instructions on how to obtain the most recent version of the data, and also includes a code snippet to automatically download the required file. We now added more information on the data source to the Appendix of the updated manuscript. Moreover, since the data is open data, we included a copy in the supplementary material.
>
> > Currently the paper does not discuss any limitations. To further understand the introduced model it would be helpful to highlight corner cases across the experiments in which the model does not perform well.
>
> The main limitation that we see is that our model treats every segment as a distinct unit. For example, if the DGP switches back and forth between two regimes,
>
> 111112222211112222211111122222,
>
> the best our model can do is to learn a segmentation into $K=6$ segments,
>
> 111112222233334444455555566666,
>
> with shared segment parameters $\theta_1 = \theta_3 = \theta_5$ and $\theta_2 = \theta_4 = \theta_6$. During training, we can favor solutions with shared segment parameters by adding a regularization term to the loss function that penalizes a high rank of the segment parameter matrix $\theta=[\theta_1,...,\theta_K]$. If we wanted to control the number of distinct regimes separately, we could learn a codebook of $C < K$ distinct segment parameters and a mapping $f$ from the segment identifiers to codebook entries $f: \\{1,...,K\\} \longrightarrow \\{1,..,C\\}$, e.g., using the vector quantization approach of [1]. We believe that this is an interesting direction for future work.
>
> [1] Aaron van den Oord, Oriol Vinyals, and Koray Kavukcuoglu. Neural discrete representation learning.
> In: NIPS, 2017.
>
> > The abstract is somewhat difficult to comprehend and appears more cryptic than necessary.
>
> Could you point us to the parts/sentences in the abstract that are difficult to follow?

---

> > ### Comment · AnonReviewer1 · 2020-11-21
> > **Response to revision**
> >
> > Thank you for addressing my comments. I would appreciate if a discussion of the limitation along the provided example (switching between two regimes) could be added to the paper or appendix/supmat.
> >
> > My initial impression w.r.t. the abstract was that it is not immediately clear what is meant by 'segmented models'. It could help to provide examples of the modeled data here. But after re-reading the paper I am less concerned about the abstract now. IMO, it can stay as is.

---

> > > ### Author Response · Authors · 2020-11-23
> > > **Limitations added**
> > >
> > > Thank you for the response! We have added a discussion of the limitations to the end of the Conclusions section. In addition to the limitation mentioned earlier in this thread, we have added a comment on model selection for the number of segments $K$.

---

### Decision · Program_Chairs · 2021-01-07
**Final Decision**

**Decision:**

Accept (Poster)

**Comment:**

In this paper, a method to solve the segmentation problem by continuous optimization is proposed by using a soft differentiable warping function. The proposed method is theoretically sound, and interesting experiments such as the data analysis of covid19 are also presented. This is a good paper in terms of both theory and application.